# Quantification of frequency-dependent genetic architectures in 25 UK Biobank traits reveals action of negative selection

Armin P. Schoech [1,2,3], Daniel M. Jordan [4], Po-Ru Loh [3,5], Steven Gazal [1,3], Luke J. O'Connor [1,2,3], Daniel J. Balick[5,6], Pier F. Palamara [7], Hilary K. Finucane [3], Shamil R. Sunyaev [3,5,6] & Alkes L. Price [1,2,3]

Understanding the role of rare variants is important in elucidating the genetic basis of human disease. Negative selection can cause rare variants to have larger per-allele effect sizes than common variants. Here, we develop a method to estimate the minor allele frequency (MAF) dependence of SNP effect sizes. We use a model in which per-allele effect sizes have variance proportional to $[p(1-p)]^{\alpha}$, where $p$ is the MAF and negative values of $\alpha$ imply larger effect sizes for rare variants. We estimate $\alpha$ for 25 UK Biobank diseases and complex traits. All traits produce negative $\alpha$ estimates, with best-fit mean of −0.38 (s.e. 0.02) across traits. Despite larger rare variant effect sizes, rare variants (MAF < 1%) explain less than 10% of total SNP-heritability for most traits analyzed. Using evolutionary modeling and forward simulations, we validate the $\alpha$ model of MAF-dependent trait effects and assess plausible values of relevant evolutionary parameters.

[1] Department of Epidemiology, Harvard T.H. Chan School of Public Health, Boston 02115 MA, USA. [2] Department of Biostatistics, Harvard T.H. Chan School of Public Health, Boston 02115 MA, USA. [3] Program in Medical and Population Genetics, Broad Institute of MIT and Harvard, Cambridge 02142 MA, USA. [4] Charles R. Bronfman Institute for Personalized Medicine, Icahn School of Medicine at Mount Sinai, New York 10029 NY, USA. [5] Division of Genetics, Department of Medicine, Brigham and Women's Hospital and Harvard Medical School, Boston 02115 MA, USA. [6] Department of Biomedical Informatics, Harvard Medical School, Boston 02115 MA, USA. [7] Department of Statistics, University of Oxford, Oxford OX1 3LB, UK. Correspondence and requests for materials should be addressed to A.P.S. (email: aschoech@hsph.harvard.edu) or to A.L.P. (email: aprice@hsph.harvard.edu)

The contribution of rare variants to the genetic architecture of human diseases and complex traits is a question of fundamental interest, which can inform the design of genetic association studies and shed light on the action of negative selection[1,2]. Recently, several studies have investigated the relationship between minor allele frequency (MAF) and trait effects[3–6]. However, these studies have analyzed a small number of traits and have not evaluated the genome-wide contribution of rare variants (MAF < 1%), which remains unknown[7].

Here we develop a profile likelihood-based mixed model method to infer MAF-dependent architectures from genotype and phenotype data. We apply our method to 25 complex traits and diseases from the UK Biobank data set, analyzing data from 113,851 individuals and 11,062,620 SNPs, including rare variants (MAF > 0.07%). Our analysis shows that rare variants have significantly increased per-allele effect sizes for most traits, with significant heterogeneity across traits. For each of these traits we also estimate the phenotypic variance explained by variants in different frequency ranges, including rare variants.

It is widely believed that frequency-dependence of SNP effect sizes is due to increased negative selection on variants that affect complex traits[1,2,8–11]. Specifically, if SNPs that affect a trait are more likely to be under negative selection, they will be enriched in the lower-frequency spectrum, so that lower-frequency SNPs will on average have larger trait effects. Thus, MAF-dependent architectures estimated from genotype and phenotype data can shed light on evolutionary parameters. Previous studies have used MAF-dependent architectures or related information to estimate a coupling parameter[9] between fitness effects and their trait effects for prostate cancer[6] and type 2 diabetes[12,13]. In this work, we use evolutionary modeling and forward simulations to investigate whether our parameterization of MAF-dependent effects ($\alpha$ model; see below) is consistent with evolutionary models, estimate the coupling between fitness effects and trait effects, and draw inferences about the average genome-wide strength of negative selection.

## Results

**Overview of methods**. We assume a previously proposed random-effect model[14,15] (here referred to as the $\alpha$ model), in which the per-allele trait effect $\beta$ of a SNP depends on its MAF $p$ via:

$$E\left(\beta^2 | p\right) = \sigma_{g,\alpha}^2 \cdot [2p(1-p)]^\alpha \tag{1}$$

A negative value of $\alpha$ implies that lower-frequency SNPs have larger per-allele effect sizes, whereas $\alpha = 0$ implies no dependence, and $\sigma_{g,\alpha}^2$ is the component of SNP effect variance that is independent of frequency. We note that Eq. (1) pertains to genome-wide SNPs, including SNPs that do not affect the trait. The $\alpha$ model is simple and convenient, but has not previously been validated by evolutionary modeling.

For a given set of genotype and phenotype data, we estimate $\alpha$ using a linear mixed model framework[16]. The model likelihood depends on $\alpha$, $\sigma_{g,\alpha}^2$, and the environmental variance (see Methods). We compute the profile likelihood over values of $\alpha$ by maximizing the likelihood with respect to $\sigma_{g,\alpha}^2$ and the environmental variance for a given $\alpha$. Our estimate $\hat{\alpha}$ is defined as the mode of the profile likelihood curve, whose width is used to compute error estimates. We show that the corresponding values of $\hat{\sigma}_{g,\alpha}^2$ can be used to estimate the SNP-heritability $h_g^2$ while accounting for MAF-dependent SNP effects, which can bias $h_g^2$ estimates when not accounted for[14,15]. We include linkage disequilibrium (LD)-dependent SNP weights[17] in our model, to avoid biases due to LD-dependent architectures[4,14,18,19]. Details of the method are described in the Methods section; we have released open-source software implementing the method (see URLs).

**Simulations**. We evaluated our method using simulations based on imputed UK Biobank genotypes[20] and simulated phenotypes, using $N = 5000$ individuals and $M = 100,000$ consecutive SNPs from a 25 Mb block of chromosome 1 (see Methods). We used default parameter settings of $\alpha = -0.3$, $h_g^2 = 0.4$, 1% of SNPs causal, imputation noise based on actual imputed genotype probabilities, and LD-dependent effects[17], but we also considered other parameter settings for each of these. Imputation noise was introduced by randomly sampling the genotypes used to simulate phenotypes from imputed genotype probabilities, while still using the expected dosage values for inference (see Methods).

In Table 1, we report $\alpha$ estimates at default and other parameter settings, both using LD-dependent weights ($\hat{\alpha}$) and without using LD-dependent weights ($\hat{\alpha}_{noLD}$). In simulations with LD-dependent effects, $\hat{\alpha}$ was unbiased at all parameter settings tested, while $\hat{\alpha}_{noLD}$ was upward biased by approximately 0.1. In simulations without LD-dependent effects, $\hat{\alpha}$ was downward biased by less than 0.1, while $\hat{\alpha}_{noLD}$ was unbiased. These simulations suggest that our method provides unbiased estimates of $\alpha$ when LD is correctly modeled, and only modestly biased estimates of $\alpha$ when LD is not correctly modeled. Importantly, imputation noise does not induce bias, indicating that our method is unbiased even if causal SNPs are not perfectly tagged. Although values of $N$ and $M$ in our main simulations were chosen to approximately match the power of the UK Biobank traits analyzed (which scales with $N/\sqrt{M}$; ref. [21]), similar results were obtained when $N$ was reduced to 2500, a setting with lower power (Table 1). Finally, we compared our profile likelihood standard error estimates to empirical standard errors from simulations. We

### Table 1 Estimates of $\alpha$ in simulations

| $\alpha$ | $h_g^2$ | Sample size | Polygenicity (%) | Imputation noise | LD dependent effects | Mean $\hat{\alpha}$ | Mean $\hat{\alpha}_{noLD}$ |
|---|---|---|---|---|---|---|---|
| −0.3 | 0.4 | 5000 | 1 | Yes | Yes | −0.276 ± 0.017 | −0.192 ± 0.019 |
| 0.0 | 0.4 | 5000 | 1 | Yes | Yes | 0.021 ± 0.020 | 0.120 ± 0.017 |
| −0.6 | 0.4 | 5000 | 1 | Yes | Yes | −0.573 ± 0.014 | −0.471 ± 0.015 |
| −0.3 | 0.2 | 5000 | 1 | Yes | Yes | −0.260 ± 0.024 | −0.148 ± 0.024 |
| −0.3 | 0.4 | 5000 | 100 | Yes | Yes | −0.308 ± 0.012 | −0.195 ± 0.013 |
| −0.3 | 0.4 | 5000 | 1 | No | Yes | −0.304 ± 0.016 | −0.191 ± 0.017 |
| −0.3 | 0.4 | 5000 | 1 | Yes | No | −0.373 ± 0.017 | −0.284 ± 0.017 |
| −0.3 | 0.4 | 2500 | 1 | Yes | Yes | −0.269 ± 0.026 | −0.157 ± 0.025 |
| −0.3 | 0.2 | 2500 | 1 | Yes | Yes | −0.266 ± 0.052 | −0.160 ± 0.034 |

We simulated phenotypes using imputed UK Biobank genotypes and applied our method to infer $\alpha$. In each line we show results from phenotypes that were simulated using various values of $\alpha$, $h_g^2$, sample size, and the proportion of causal SNPs. In most simulations, imputation noise and LD dependent SNP effects were included in the simulated phenotypes. In each case we report the mean estimated $\alpha$ and standard error of the mean, using our estimation method either with LD correction ($\hat{\alpha}$) or without LD correction ($\hat{\alpha}_{noLD}$).

determined that standard error estimates slightly underestimated true standard errors at $N = 5000$, $M = 100{,}000$ (likely due to the small number of causal SNPs in these simulations), but were well-calibrated in simulations at larger values of $N$ and $M$ (see Supplementary Table 1), indicating well-calibrated error estimates when using even larger values of $N$ and $M$ in our analysis of UK Biobank traits. The profile likelihood curves were smooth and unimodal at all parameter settings (see Supplementary Figure 1).

Although the main focus of this paper is on obtaining and interpreting estimates of $\alpha$, we also used our simulation framework to evaluate the effectiveness of our method in obtaining SNP-heritability estimates that avoid biases due to MAF-dependent and LD-dependent architectures. In Supplementary Table 2 we report SNP-heritability estimates using our method, both using LD-dependent weights ($\hat{h}_\alpha^2$) and without using LD-dependent weights ($\hat{h}_{\alpha,\mathrm{noLD}}^2$), and using GCTA with a single variance component ($\hat{h}_{\mathrm{GCTA}}^2$)[22]. $\hat{h}_\alpha^2$ and $\hat{h}_{\alpha,\mathrm{noLD}}^2$ were roughly unbiased at all parameter settings, while GCTA with a single variance component produced biased estimates, consistent with previous work[4,14]. Other methods of avoiding bias due to MAF-dependent and LD-dependent architectures have recently been proposed, including GREML-LDMS[4] and LDAK[19]; a complete benchmarking of SNP-heritability estimation methods is provided in ref.[23].

**Analysis of 25 UK Biobank traits.** We applied our method to 113,851 British-ancestry individuals from the UK Biobank with 1000 Genomes- and UK10K-imputed genotypes at 11,062,620 SNPs with at least 5 minor alleles in the UK10K reference panel (MAF > 0.07%; see Methods). We analyzed 25 heritable, polygenic traits with at least 50% of individuals phenotyped (Table 2). Phenotype values were corrected for fixed effects, including sex and 10 principal components (see Methods). Profile likelihood curves for all 25 traits are displayed in Supplementary Figure 2. We observed that the curves were smooth and unimodal (consistent with simulations; Supplementary Figure 1), suggesting that estimates of $\alpha$ are likely to be robust.

In Table 2, we report estimates of $\alpha$ for all 25 traits. All traits had negative $\alpha$ estimates (with most estimates lying between $-0.5$ and $-0.2$), and 20 traits had significantly negative estimates (i.e., 95% credible intervals did not overlap zero), implying that lower-frequency SNPs have larger per-allele effect sizes. We observed statistically significant heterogeneity in estimates of $\alpha$ across the 25 traits ($P = 0.0014$), consistent with different levels of (direct and/or pleiotropic) negative selection across traits (see Discussion); our test for heterogeneity accounts for estimation noise, which varies across traits depending on heritability and sample size (see Methods). We estimated the underlying distribution of true (unobserved) values of $\alpha$ to have mean $-0.38$ (s.e. 0.02) and standard deviation 0.08 (s.e. 0.03), assuming a normal distribution (see Methods). We obtained very similar results when repeating the entire analysis using 9,336,687 SNPs with MAF > 0.3% in the UK10K reference panel (Supplementary Table 3); we note that these results are unlikely to be affected by imputation error, because simulation results in Table 1 show that our method is not significantly affected by imputation error under correctly calibrated imputation accuracies, and because we further determined that MAF > 0.3% SNPs generally have well-calibrated imputation accuracies (Supplementary Figure 3).

We estimated the proportion of SNP-heritability explained by SNPs in each part of the MAF spectrum, for different values of $\alpha$. This computation relies on the empirical MAF spectrum in UK10K, as heritability per MAF bin depends both on heritability per SNP and number of SNPs per MAF bin (see Methods). Results are reported in Fig. 1. We determined that rare and low-

| Phenotype | Sample size | $\hat{\alpha}$ [95% CI] |
|---|---|---|
| Age of menarche | 58,329 | −0.40 [−0.63, −0.11] |
| Blood pressure (diastolic) | 104,835 | −0.39 [−0.54, −0.20] |
| Blood pressure (systolic) | 104,835 | −0.38 [−0.54, −0.18] |
| BMI | 113,540 | −0.24 [−0.38, −0.06] |
| Bone mineral density | 110,611 | −0.35 [−0.45, −0.23] |
| FEV1/FVC | 97,075 | −0.44 [−0.55, −0.31] |
| FVC | 97,075 | −0.15 [−0.31, 0.04] |
| Height | 113,660 | −0.45 [−0.52, −0.39] |
| Smoking status | 113,560 | −0.16 [−0.43, 0.21] |
| Waist-hip ratio | 113,668 | −0.17 [−0.43, 0.19] |
| Allergic eczema | 113,707 | −0.60 [−0.85, −0.26] |
| Asthma | 113,707 | −0.25 [−0.60, 0.28] |
| College education | 112,811 | −0.32 [−0.54, −0.04] |
| Hypertension | 113,689 | −0.18 [−0.46, 0.21] |
| Eosinophil count | 108,957 | −0.40 [−0.54, −0.24] |
| High light scatter reticulocyte count | 108,785 | −0.53 [−0.65, −0.38] |
| Lymphocyte count | 108,664 | −0.52 [−0.63, −0.38] |
| Mean corpuscular hemoglobin | 108,513 | −0.42 [−0.53, −0.31] |
| Mean sphered cell volume | 109,523 | −0.43 [−0.56, −0.28] |
| Monocyte count | 110,026 | −0.19 [−0.35, −0.01] |
| Platelet count | 109,971 | −0.19 [−0.32, −0.03] |
| Platelet distribution width | 109,938 | −0.27 [−0.44, −0.07] |
| Red blood cell count | 110,054 | −0.39 [−0.51, −0.25] |
| Red blood cell distribution width | 109,913 | −0.20 [−0.36, −0.01] |
| White blood cell count | 110,186 | −0.25 [−0.42, −0.03] |

We computed $\alpha$ estimates for 25 UK Biobank traits, including 10 quantitative traits, 4 case–control traits, and 11 blood cell traits (all quantitative). The reported 95% credible intervals were calculated from the profile likelihood curves using a flat prior

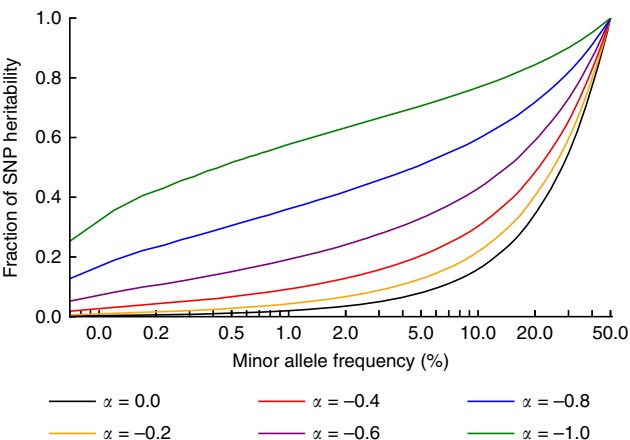

**Fig. 1** Fraction of SNP-heritability in different MAF ranges given $\alpha$. We report the fraction of SNP-heritability explained by SNPs up to a certain MAF (x-axis), for different values of $\alpha$. For example, assuming $\alpha = -0.4$, SNPs with MAF ≤ 5% collectively explain about 20% of the total SNP-heritability. These results are based on the UK10K allele frequency spectrum and our model assumption that squared per-allele effects are proportional to $[2p(1-p)]^\alpha$. Source data are provided as a Source Data file

frequency variants contribute a very small proportion of SNP-heritability at the mean $\alpha$ estimate of $-0.38$, and a relatively small proportion of SNP-heritability even for the most negative $\alpha$ estimate of $-0.60$. Specifically, at $\alpha = -0.38$ (s.d. 0.08), only 8.9% (s.d. 2.7%) of SNP-heritability is explained by SNPs with MAF < 1%. We also used $\hat{\alpha}$ to obtain total SNP-heritability estimates corrected for biases due to MAF-dependent and LD-dependent architectures for each of the 25 traits (Supplementary Table 4; see Methods).

The above analysis used LD-dependent weights based on a model of ref. [17], with an LD-dependent architecture parameter of $\tau^* = -0.3$. (A negative value of $\tau^*$ implies that, at a given MAF, low-LD SNPs have larger causal effect sizes.) We also performed an analysis in which MAF-dependent and LD-dependent effects are estimated jointly, maximizing the profile likelihood over values of both $\alpha$ and $\tau^*$ for each of the 25 UK Biobank traits (see Methods). Results are reported in Supplementary Table 5. The best-fit $\tau^*$ was generally close to $\tau^* = -0.3$. Although a different value of $\tau^*$ provided the best fit for some traits, the overall impact on $\alpha$ estimates was small, with the best-fit distribution of true (unobserved) values of $\alpha$ across traits changing from $-0.38$ (s.d. 0.08) to $-0.35$ (s.d. 0.11).

**Effect of negative selection on MAF-dependent architectures.** Frequency-dependent trait effect sizes have been widely attributed to negative (purifying) selection on variants that affect complex traits, which causes them to be enriched for lower-frequency variants, so that lower-frequency SNPs will have larger traits effects[1,2,8–11]. Here we use evolutionary modeling to predict the frequency-dependent architecture of a trait, given the coupling between fitness effects and trait effects. The aim of this analysis was to investigate whether the $\alpha$ model (Eq. 1) is consistent with the predictions of evolutionary models, and to draw conclusions about evolutionary parameters from our estimates of $\alpha$ across 25 UK Biobank traits.

We used an evolutionary model of Eyre-Walker[9], which introduces a parameter $\tau$ quantifying the coupling between a SNP's fitness effect (selection coefficient $s$) and target trait effect size ($\beta$); $\tau > 0$ implies that SNPs under negative selection have larger trait effect sizes on average, whereas $\tau = 0$ corresponds to no coupling. Using this model, we derived two analytical results. First, it is straightforward to show that

$$\mathrm{E}\left(\beta^2 | p\right) \propto \mathrm{E}(s^{2\tau} | p) \qquad (2)$$

where $p$ is minor allele frequency (see Methods). This implies that increased trait effects for lower-frequency variants requires both that lower-frequency variants have significantly larger selection coefficients $s$ and that $\tau > 0$. Second, based on Eq. (2), we analytically evaluated $\mathrm{E}(s^{2\tau}|p)$ to quantify the MAF-dependence of SNP effects under the Eyre-Walker model (see Methods). In this derivation, we ignored LD between selected SNPs, assumed a constant effective population size $N_e$, and assumed that selection coefficients $s$ of SNP loci across the genome are drawn from a gamma distribution, with mean $\bar{s}$ and shape parameter $k$ (ref. [24]). (We note that $k$ parametrizes the polygenicity of fitness and trait effects; see Methods.) Under these assumptions, we derived the result that there exists a MAF threshold $T$ such that for $p > T$ the $\alpha$ model approximately holds, but for $p < T$ trait effects are approximately independent of frequency (see Methods). The threshold is

$$T = \frac{k}{4N_e\bar{s}} \qquad (3)$$

Intuitively, this threshold corresponds to the maximum frequency at which even the most strongly selected SNPs are

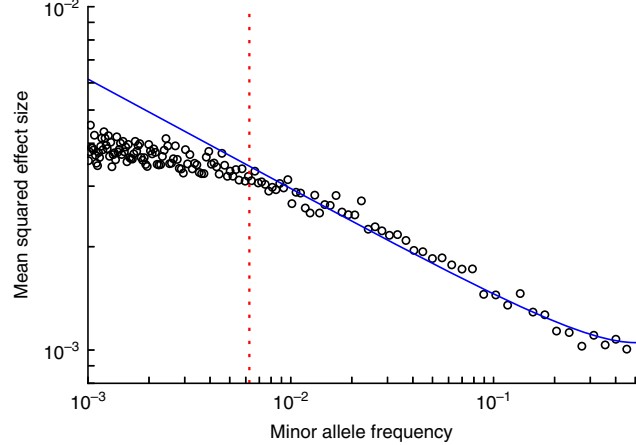

**Fig. 2** MAF-dependence of SNP effects in evolutionary forward simulations. Forward simulations confirm that $\alpha$ model approximately holds above the MAF threshold $T = \frac{k}{4N_e\bar{s}}$. We report simulated mean squared SNP effect sizes at a given MAF on a log-log plot, assuming $\tau = 0.4$ and a genome wide selection coefficient distribution with mean $\bar{s} = 10^{-3}$ and shape parameter $k = 0.25$. Data points represent the mean squared effect size of 1000 SNPs of similar MAF, calculated assuming Eq. (2). The blue curve represents mean squared effect sizes under the $\alpha$ model (Eq. 1) with $\alpha = -0.32$, fitted to SNPs above the MAF threshold $T$. The MAF threshold $T = 0.006$ is indicated by a dotted red line. Source data are provided as a Source Data file

still only affected by genetic drift, with their frequency being too low to be significantly affected by selection. We note that $T$ is independent of the trait analyzed, since $\bar{s}$ and $k$ parametrize the distribution of genome-wide selection coefficients.

Although our derivation of Eq. (3) ignored the effects of demographic changes and LD, we confirmed this result by performing forward simulations using SLiM2[25], using a European demographic model[26] and realistic LD patterns (see Methods). Specifically, for a given $\tau$ we computed $E(s^{2\tau}|p)$ in Eq. (2) from the $s$ and $p$ values of simulated SNPs. Our main simulations assumed $\tau = 0.4$, $N_e = 10,000$, $\bar{s} = 0.001$ and $k = 0.25$ (ref. [24]), so that $T = 0.006$ (Eq. 3). Results are reported in Fig. 2, which shows that for $p > T = 0.006$ the $\alpha$ model with best-fit $\alpha = -0.32$ provides a good fit, but for $p < T = 0.006$ the effect sizes are less MAF-dependent and are thus significantly smaller than expected under the $\alpha$ model. Results at other parameter settings were qualitatively similar, with the threshold varying according to Eq. (3) (see Supplementary Figure 4). These simulations provide an important validation of our analytical derivations, which are limited by unrealistic assumptions of no LD and constant population size.

We sought to draw inferences about the threshold $T$ for UK Biobank traits. To do so, we computed values of $\hat{\alpha}_{\mathrm{common}} - \hat{\alpha}$, where $\hat{\alpha}_{\mathrm{common}}$ is similar to $\hat{\alpha}$ but uses only SNPs with MAF > 5% for inference (6,273,557 SNPs with MAF > 5% instead of 11,062,620 SNPs with MAF > 0.07%). If a large proportion of SNPs had MAF below $T$, we would expect to obtain smaller (more negative) values of $\hat{\alpha}_{\mathrm{common}}$, since SNPs of MAF below $T$ with less MAF-dependent effects would be ignored. However, $\hat{\alpha}_{\mathrm{common}} - \hat{\alpha}$ was not significantly different from zero for any of the 25 UK Biobank traits (see Supplementary Table 6), nor was the best-fit estimate across traits, which actually increased slightly from $-0.38$ (s.e. 0.02) for $\hat{\alpha}$ to $-0.35$ (s.e. 0.02) for $\hat{\alpha}_{\mathrm{common}}$. We subsequently simulated traits using $T_{\mathrm{sim}}$ of 0%, 5% and 10% and genome-wide UK Biobank genotypes (see Methods), in order to assess which value of $T_{\mathrm{sim}}$ was most consistent with $\hat{\alpha}_{\mathrm{common}} - \hat{\alpha}$ for UK Biobank traits. (We caution that this should not be viewed

as a formal hypothesis test for the value of $T$.) Specifically, we simulated traits mimicking UK Biobank height in heritability, sample size and $\alpha$. Due to computational constraints, we did not simulate traits other than height; however, since $T$ is the same across traits (see Eq. 3), we expect conclusions to be similar. (We note that these simulations did not make any assumptions about demography, but simply simulated SNP effects on the target trait using the $\alpha$ model with threshold $T_{sim}$ and UK Biobank imputed genotypes.) We determined that the $\hat{\alpha}_{common} - \hat{\alpha}$ value of 0.02 (s.e. 0.05) for UK Biobank height was significantly different from the value for $T_{sim} = 0\%$ ($-0.10$ (s.d. 0.02); $p = 0.01$ for difference), and very significantly different from values for $T_{sim} = 5\%$ ($-0.16$ (s.d. 0.01); $p = 0.0002$ for difference) and $T_{sim} = 10\%$ ($-0.19$ (s.d. 0.02); $p = 0.0007$ for difference) (see Supplementary Table 7), suggesting some form of model misspecification. In simulations with more strongly LD-dependent architectures for common variants, results were concordant with UK Biobank height for $T_{sim} = 0\%$ only ($-0.01$ (s.d. 0.07); $p = 0.7$ for difference) (see Supplementary Table 7). However, other forms of model misspecification are possible, such as violations of the Eyre-Walker fitness-trait coupling model[9] (see Discussion and refs. [27,28]) or of the assumption of gamma distributed SNP fitness effects. In summary, our results are somewhat supportive of $T < 5\%$, i.e., the $\alpha$ model provides a good fit for common SNPs (MAF $\geq 5\%$) but may overestimate SNP effect magnitudes of SNPs with MAF $< 5\%$, but this should not be viewed as a conclusive finding.

Finally, we sought to draw conclusions about the values of the average genome-wide selection coefficient $\bar{s}$ and the Eyre-Walker coupling parameter $\tau$. First, a threshold $T < 5\%$ would imply an average selection coefficient $\bar{s} > \frac{5k}{N_e}$. Assuming $N_e = 10,000$ (ref. [29]) and $k = 0.25$ (ref. [24]), $\bar{s}$ would be on the order of $10^{-4}$ or stronger. We caution that this bound on $\bar{s}$ relies on our bound on $T$, which should not be viewed as a conclusive finding. Second, we determined that the best-fit estimate of $\hat{\alpha} = -0.38$ across 25 traits corresponds to a $\tau$ value in the range [0.3,0.5] (Fig. 3; see Methods). We reached this conclusion by repeating our forward simulations for $\tau \in [0,1]$ (vs. $\tau = 0.4$ above), $\bar{s} \in \{0.0001, 0.001\}$ (vs. 0.001 above) and $k \in \{0.125, 0.25\}$ (vs. 0.25 above) and fitting the $\alpha$ model using SNPs above the threshold $T$ from Eq. (3). Figure 3 shows that the best-fit $\alpha$ depends primarily on $\tau$, with only weak dependence on $\bar{s}$ and $k$. Estimates of $\tau$ for each of the 25 traits are provided in Supplementary Table 8. We caution that the above conclusions about $\bar{s}$ and $\tau$ rely on an estimate of the shape parameter $k$ from ref. [24], which focused on coding SNPs. It is possible that $k$ may differ substantially between coding and non-coding variants. We thus repeated our analyses at a wider range of values of $k$ (0.0625, 0.125, 0.25, 0.5, and 1). First, a value of $k \geq 0.0625$ would imply that $\bar{s}$ is likely on the order of $3 \times 10^{-5}$ or stronger. Second, the best-fit estimate of $\hat{\alpha} = -0.38$ across 25 traits now corresponds to a $\tau$ value in the range [0.3,0.8] (Supplementary Figure 5).

## Discussion

We have quantified the MAF-dependent architectures of 25 diseases and complex traits under the $\alpha$ model[14,15] (Eq. 1). We inferred negative values of $\hat{\alpha}$ for all 25 traits and significantly negative values for 20 traits, corresponding to higher trait effects for lower-frequency SNPs. The best-fit distribution of $\alpha$ across traits had mean $-0.38$ (s.e. 0.02) and standard deviation 0.08 (s.e. 0.03), implying that only 8.9% (s.d. 2.7%) of SNP-heritability is explained by rare SNPs (MAF $< 1\%$), despite significantly larger effects for rare variants. Although rare variants explain relatively little heritability, rare variant association studies may still identify variants of large effect that reveal interesting biology and

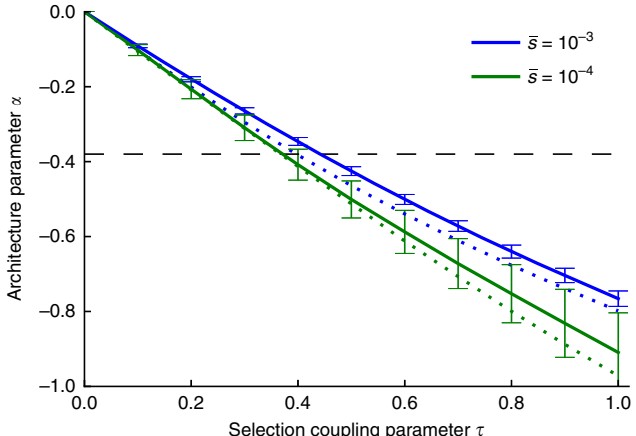

**Fig. 3** Value of $\alpha$ as a function of $\tau$ and other parameters in forward simulations. We report best-fit $\alpha$ estimates for simulations at each value of $\tau$ at a given genome-wide average selection coefficient $\bar{s}$. Selection coefficients were sampled using a gamma distribution shape parameter of $k = 0.25$ (solid lines) or $k = 0.125$ (dotted lines). $\alpha$ estimates where calculated by fitting the model in Eq. (1) to simulated SNP effects above twice the MAF threshold $2T = \frac{k}{2N_e\bar{s}}$ (in order to avoid edge effects near $T$), with error bars representing standard errors calculated by bootstrap resampling of 25 independent SLiM2 simulations. The horizontal dashed line indicates $\alpha = -0.38$, the best-fit $\alpha$ across the 25 UK Biobank traits. Results for a broader range of $k$ values are reported in Supplementary Figure 5. Source data are provided as a Source Data file

actionable drug targets[11,30]. On the other hand, rare variants will likely play only a limited role in polygenic risk prediction, which will be largely driven by common variants.

Using evolutionary modeling and simulations, we determined that our results are somewhat supportive of $T < 5\%$, i.e., the $\alpha$ model provides a good fit for common SNPs (MAF $\geq 5\%$) but may overestimate effect magnitudes of rare and low-frequency SNPs; our estimate of 8.9% (s.d. 2.7%) of SNP-heritability explained by rare SNPs (MAF $< 1\%$) should therefore be viewed as a suggestive upper bound. This would imply an average genome-wide negative selection coefficient on the order of $10^{-4}$ or stronger, given the MAF-dependent architectures that we inferred. The best-fit $\alpha$ estimate across 25 traits implies an Eyre-Walker[9] $\tau$ parameter between 0.3 and 0.5, quantifying the coupling between fitness effects and trait effects. These findings are conditional on the assumption that the shape parameter $k$ of the fitness effect distribution of genome-wide SNPs does not substantially differ from the value inferred for coding variants by ref. [24]. Under a broader range of possible values of $k$, $\bar{s}$ would be on the order of $3 \times 10^{-5}$ or stronger, and the best-fit $\alpha$ estimate across 25 traits implies a $\tau$ value between 0.3 and 0.8. We caution that these results may be impacted by violations of model assumptions, such as the Eyre-Walker fitness-trait coupling model[9] (see discussion below) or the gamma distribution of SNP fitness effects; these assumptions have been frequently employed[6,9,12,13], but may not perfectly fit UK Biobank traits. Our finding that estimates of $\alpha$ (and hence $\tau$) vary only modestly across traits is consistent with the action of pleiotropic selection, in which SNPs that affect the target trait also affect other selected traits[27,31]; under direct selection, greater variation in $\tau$ would be expected, and traits that are not directly selected would have $\tau = 0$.

Recent studies have investigated MAF-dependent architectures in genome-wide analyses of schizophrenia[3,5], as well as height and BMI[4]. These studies analyzed a small number of traits, and either did not analyze rare variants[3,5] or aggregated all MAF $< 10\%$ variants into a single MAF bin[4], underscoring the difficulty

of obtaining precise estimates of rare variant heritability using the MAF bin approach. Another study used targeted sequencing of 63 prostate cancer risk regions to conclude that 42% (s.e. 11%) of the prostate cancer SNP-heritability attributable to these regions in African Americans is due to rare SNPs (MAF < 1%), although rare variant heritability in Europeans was non-significant[6].

A more recent study introduced a revised LDAK method[19] (revising an earlier LDAK method[14]) and estimated a parameter that it referred to as $\alpha$. We refer to this parameter as $\alpha_{LDAK}$, because it is different from the parameter $\alpha$ that was previously described in refs. [14,15] and that is defined and estimated in this paper. Specifically, the Discussion section of ref. [19] states that the SNP effect size variance is proportional to $[p_j(1-p_j)]^{\alpha_{LDAK}}$. However, that statement is incorrect. Actually, under the model of ref. [19], the SNP effect size variance is proportional to $[p_j(1-p_j)]^{\alpha_{LDAK}} \cdot w_j$, where $w_j$ is an LD-dependent weight (see Eq. (1) of ref. [19]). Unlike the LD-dependent weights that we use[17], $w_j$ is dependent on MAF, with lower frequency SNPs having higher values of $w_j$. Thus, SNP effect size is specifically not proportional to $[p_j(1-p_j)]^{\alpha_{LDAK}}$, and $\alpha_{LDAK}$ is a parameter that is different from $\alpha$. Indeed, our simulations confirmed that estimates of $\alpha_{LDAK}$ obtained using the LDAK software were upward biased by roughly 0.4 compared to the true $\alpha$ as defined in previous work[14,15] and this paper (see Supplementary Table 9). Thus, the revised LDAK method and software[19] cannot be used to estimate $\alpha$.

A study conducted in parallel to this work investigated MAF-dependent architectures of 28 UK Biobank traits[32] using a Bayesian method to estimate a parameter identical to the $\alpha$ parameter that we estimate. Results of ref. [32] were broadly similar to our results, but we note four key differences between the studies. First, ref. [32] analyzed 483,634 Affymetrix SNPs with MAF > 1%, noting that analyzing a larger number of imputed SNPs would be computationally challenging for their BayesS inference model. On the other hand, we were able to analyze 11,062,620 typed and imputed SNPs with MAF > 0.07%, because the most computationally intensive step of our method (running REML; see Methods) is independent of the number of SNPs. However, we determined here that inclusion or exclusion of rare variants does not significantly affect our results (see Supplementary Table 6). Second, ref. [32] used an elegant approach to infer the polygenicity of each trait. Third, ref. [32] estimated SNP effects jointly and conditionally using a sparse model, which accounts for LD between SNPs but does not account for LD-dependent causal effect sizes[17]. Our method accounts for LD-dependent causal effects, and we show that this has a non-negligible impact on our estimates of $\alpha$ (see Table 1); it is currently unclear whether estimates of ref. [32] would be impacted by LD-dependent causal effect sizes. Fourth, although ref. [32] performed forward simulations to show that their findings implicate negative selection on trait-affecting SNPs, they did not use these simulation results to investigate the validity of their parametric inference model or to investigate evolutionary parameters. We note that both ref. [32] and our method rely on the $\alpha$ model (Eq. 1), a 1-parameter model of the relationship between allele frequencies and trait effect sizes. Evaluating the full joint distribution of allele frequencies and trait effect sizes could yield interesting additional information, but would require new approaches that might be extremely computationally intensive when applied to very large data sets.

In addition, several recent studies have drawn inferences about evolutionary parameters that affect complex traits. Refs. [12,13], estimated $\tau$ in type 2 diabetes to be approximately 0.1, by comparing the number of rare and low-frequency associations in empirical

studies to the number in simulations. Ref. [6] estimated $\tau$ by matching the heritability explained by rare SNPs (MAF < 1%) in their analysis of prostate cancer to simulation results, inferring $\hat{\tau} = 0.48$ (95% CI [0.19,0.78]). We are not aware of any previous study that investigated plausible values of the genome-wide average strength of negative selection, although ref. [27] used a different modeling approach to estimate the mutational target load.

We note several limitations in our work. First, our analyses are restricted to high-prevalence diseases and quantitative traits, as low-prevalence diseases are not well-represented in the UK Biobank due to random ascertainment. This motivates additional analyses of low-prevalence diseases, which could potentially be subject to stronger direct selection. However, we caution that our method might be susceptible to biases when used to analyze ascertained case-control traits, as previously described for linear mixed model based heritability estimation methods[33,34], meriting further investigation. Second, we use the Eyre-Walker model[9] to parameterize the coupling between fitness effects and trait effects. The Eyre-Walker model has previously proven useful in a variety of settings[6,12,13], but other coupling models are also possible[27,28] and merit further exploration. One limitation of the Eyre-Walker model is that it does not allow for signed correlations between SNP trait effect and selection coefficient, i.e., the damaging allele is equally likely to reduce or increase the trait value. This assumption is violated when the target trait is under direct selection, but is plausible if selection on the SNP is mainly pleiotropic, which appears to be the dominant form of selection for the traits analyzed here (see above). Third, we assume that the distribution of selection coefficients follows a gamma distribution. This assumption implies that there are no outlier SNPs under exceptionally strong negative selection. Such extremely selected SNPs would stay at very low frequencies and only affect our results if they had extreme effects on the target trait. However, such SNPs have not been identified for most complex traits[2]. Fourth, our analytic derivations ignore LD and assume a constant population size. Our derivations imply that $\alpha = -2\tau$ (see Methods), but our forward simulations, which include realistic LD patterns and demography, suggest that $\alpha = -\tau$. The direction of this change is consistent with the action of background selection due to LD, since strong LD leads to a SNP's frequency being influenced not only by its own selection coefficient but also by the selection coefficients of many other correlated SNPs, leading to a less negative $\alpha$ value for a given $\tau$. However, this difference could potentially also be due to demography[35]. The impact of LD and demography on $\alpha$ could potentially be investigated further using forward simulations. Finally, our forward simulations assume that negative (purifying) selection is the dominant mode of selection affecting complex traits. Although positive selection is likely to affect some loci, recent work has suggested that selective sweeps were rare in human evolution[36] and hence unlikely to have substantial genome-wide effects on MAF-dependent trait architectures. We also did not investigate the potential effects of stabilizing selection[27]. Despite these limitations, our quantification of MAF-dependent effect sizes and investigation of the underlying evolutionary parameters is broadly informative for the genetic architectures of diseases and complex traits.

## Methods

**Inferring frequency dependence of SNP effects.** We assume a linear complex trait model for $N$ individuals and $M$ SNPs with

$$\mathbf{y} = \mathbf{X}\boldsymbol{\beta} + \boldsymbol{\varepsilon}, \text{ with } \varepsilon_i \sim N(0, \sigma_\varepsilon^2) \text{ i.i.d.} \tag{4}$$

Here, $\mathbf{y}$ is a vector of $N$ phenotype values with mean zero, $\mathbf{X}$ is the mean-centered genotype matrix, $\boldsymbol{\beta}$ is the vector of $M$ SNP effects and $\boldsymbol{\varepsilon}$ is a vector of environmental effects (i.e., any non-SNP effects). Furthermore we assume the effect

size of SNP $j$ to be a random variable that follows a distribution depending on its minor allele frequency (MAF) $p_j$:

$$\beta_j \sim N\left(0, \sigma_{g,\alpha}^2 \cdot \left[2p_j\left(1 - p_j\right)\right]^{\alpha}\right), \qquad (5)$$

where effect sizes of two SNPs are independent conditional on their allele frequencies. A negative $\alpha$ value indicates larger trait effects on average for lower-frequency SNPs, whereas $\sigma_{g,\alpha}^2$ is the component of the SNP effect variance independent of frequency. This model, which we call the $\alpha$ model, has been used in previous analyses of complex traits[14,15]. We note that $\beta$ defines the per-allele SNP effect which is distinct from the heritability explained by a SNP. Under Hardy-Weinberg equilibrium and given Eq. (5), the average heritability explained by a SNP of frequency $p$ is proportional to $[2p(1-p)]^{1+\alpha}$.

From Eqs. (4) and (5) it follows that the distribution of the phenotype vector $\mathbf{y}$ is a multivariate normal distribution with

$$\mathbf{y} \sim N_N\left(0, \mathbf{X}\mathbf{D}_{\alpha}\mathbf{X}^T\sigma_{g,\alpha}^2 + \mathbf{I}\sigma_{\varepsilon}^2\right), \mathbf{D}_{\alpha} \text{ diagonal with } (\mathbf{D}_{\alpha})_{jj} = \left[2p_j\left(1 - p_j\right)\right]^{\alpha} \quad (6)$$

Given the genotype matrix $\mathbf{X}$, SNP frequency vector $\mathbf{p}$ and phenotype vector $\mathbf{y}$, the likelihood over the three parameters $\sigma_{g,\alpha}^2$, $\sigma_{\varepsilon}^2$ and $\alpha$ is fully defined by Eq. (6). Hence, the MLE of the parameter triple $(\sigma_{g,\alpha}^2, \sigma_{\varepsilon}^2, \alpha)$ can be found directly by maximizing the corresponding likelihood. Since we are primarily interested in estimating $\alpha$, we used a profile likelihood based approach, with the profile likelihood of $\alpha$ defined as $L_{\mathrm{prof}}(\alpha) = \max_{(\sigma_{g,\alpha}^2, \sigma_{\varepsilon}^2)} L(\sigma_{g,\alpha}^2, \sigma_{\varepsilon}^2, \alpha)$. In this analysis we use $\hat{\alpha} = \mathrm{argmax}_{\alpha} L_{\mathrm{prof}}(\alpha)$ as the estimator of $\alpha$, given genotype and phenotype data $\mathbf{X}$ and $\mathbf{y}$. $\hat{\alpha}$ is also equal to the $\alpha$ value in $(\sigma_{g,\alpha}^2, \sigma_{\varepsilon}^2, \alpha)$ that maximizes the total likelihood in Eq. (6).

In practice, the profile likelihood $L_{\mathrm{prof}}(\alpha)$ was derived in the following way: for some $\alpha'$, $\mathbf{X}\mathbf{D}_{\alpha'}\mathbf{X}^T$ was calculated. Given phenotype values $\mathbf{y}$ and for a given $\alpha'$, we inferred maximum likelihood estimates for $\sigma_{g,\alpha}^2$ and $\sigma_{\varepsilon}^2$ via restricted maximum likelihood estimation[37], using the GCTA software implementation[37]. This procedure was repeated for a range of $\alpha'$. Here we used a minimal range of $\alpha' \in \{-1.00, -0.95, \ldots, 0.00\}$ for all traits, but extended the range to higher values if necessary, such that there is a minimal difference of 5 in log profile likelihood between the mode and the boundary. This ensures that the part of the curve that is significantly above zero is sampled. These data points were then interpolated with a natural cubic spline, yielding the final profile likelihood curve. Credible intervals for $\hat{\alpha}$ were estimated by combining the profile likelihood curve with a flat prior. Although our above modeling assumes a quantitative trait, this method is equally applicable to randomly ascertained case–control traits since all likelihood calculations are performed using the GCTA software, which analyzes case–control traits accordingly via a liability threshold model[38].

Given $\hat{\alpha}$ for a set of phenotypes, the cross-trait estimate, $\hat{\alpha}_{\mathrm{cross-trait}}$, was calculated as the inverse variance weighted mean across the traits. We tested for heterogeneity of true underlying $\alpha$ values across $n$ traits by comparing $\sum_{i=1}^{n} \frac{(\hat{\alpha}_i - \hat{\alpha}_{\mathrm{cross-trait}})^2}{\mathrm{std.error}_i^2}$ to a $\chi_n^2$ null statistic. The best-fit standard deviation in true $\alpha$ values across traits, was calculated by assuming normally distributed true $\alpha$ with mean $\hat{\alpha}_{\mathrm{cross-trait}}$, and then choosing the standard deviation, for which the variance of the simulated $\hat{\alpha}$ using the inferred standard errors matched the variance of the 25 $\alpha$ estimates most closely. We note that by accounting for the standard errors in our estimates of $\alpha$, our approach ensures that any heterogeneity that is detected reflects true differences in frequency-dependent architectures and not merely differences due to estimation noise, which varies across traits depending on heritability and sample size.

**Correcting for LD-dependent architectures.** Ref. [17] showed that for a given MAF, SNPs with higher LD have lower per-allele effects on average. Specifically, they use level of LD (LLD), defined as the rank-based inverse normal transform of the LD score. LLD is transformed separately in each part of the MAF spectrum, ensuring that it is independent of MAF. Ref. [17] reported that SNPs that have LLD one standard deviation above the mean have a squared per-allele effect size reduced by $(30 \pm 2)\%$ on average. This violates our assumption that, at a given MAF, all SNP effects are independent and identically distributed.

To avoid bias in our estimation due model misspecification, we incorporated LD-dependent SNP effects by changing Eq. (5) to

$$\beta_j | p_j, \mathrm{LLD}_j \sim N\left(0, \sigma_{g,\alpha}^2 \cdot \left[2p_j\left(1 - p_j\right)\right]^{\alpha} \cdot \left(1 - 0.3 \cdot \mathrm{LLD}_j\right)\right) \qquad (7)$$

This expression incorporates the LD dependence of ref. [17], however, since LLD has mean zero and is independent of MAF, $\beta_j | p_j \sim N\left(0, \sigma_{g,\alpha}^2 \cdot \left[2p_j\left(1 - p_j\right)\right]^{\alpha}\right)$ still holds, even though effect sizes $\beta$ are not i.i.d. given $p$. To remove the LD dependence in the effect size distribution, we calculated a renormalized genotype matrix $\tilde{\mathbf{X}}$, with $\tilde{X}_{ij} = X_{ij} \cdot \left(1 - 0.3 \cdot \mathrm{LLD}_j\right)^{1/2}$. This effectively changes the complex trait model in Eq. (4) to $\mathbf{y} = \tilde{\mathbf{X}}\beta + \varepsilon$, where now $\tilde{\beta}_j | p_j \sim$

$N\left(0, \sigma_{g,\alpha}^2 \cdot \left[2p_j\left(1 - p_j\right)\right]^{\alpha}\right)$ is again i.i.d. for a fixed $p$. Unless otherwise stated, we hence estimated $\alpha$ using $\tilde{\mathbf{X}}$ instead of $\mathbf{X}$ to avoid biases due to LD-dependent architectures.

When estimating LD-dependent architectures, we used an LD-weight parameter $\tau^*$, similar to ref. [17]. We assume that

$$\beta_j | p_j, \mathrm{LLD}_j \sim N\left(0, \sigma_{g,\alpha}^2 \cdot \left[2p_j\left(1 - p_j\right)\right]^{\alpha} \cdot \left(1 + \tau^* \cdot \mathrm{LLD}_j\right)\right) \qquad (8)$$

We jointly estimated $\alpha$ and $\tau^*$ by identifying the pair of values $(\alpha, \tau^*)$ that maximizes the profile likelihood using a 2D grid, with $\tau^* = -0.60, -0.45, -0.30, -0.15, 0.00$ and $\alpha = -1.00, -0.95, -0.90, \ldots$ (the same set of $\alpha$ values that we analyzed previously).

**Genotype data.** We use the UK Biobank phase 1 data release (see URLs), which comprises of data from 152,729 individuals genotyped at 847,131 SNP loci. Here, we only used data from 113,851 individuals following selection criteria previously used by ref. [39]: individuals were selected to have self-reported and confirmed British ancestry and related individuals were removed from the analysis such that the pairwise genetic relatedness is < 5% (after LD-pruning SNPs). Individuals that had withdrawn consent to participate in the UK Biobank project after initial publication were removed from the analysis. We used imputed genotype data as provided by UK Biobank. These genotypes were imputed using the IMPUTE2 software[40] and a joint reference panel from the UK10K project[41] and 1000 Genomes Phase 3 (ref. [42]). The resulting imputed genotype data includes roughly 70,000,000 SNP loci across the 22 autosomal chromosomes. The data was downloaded in the BGEN file format (see URLs), a compressed file format that includes—for each individual and variant site—the probability of being homozygous reference, heterozygous, or homozygous alternative. Due to imputation uncertainty, the genotype matrix $\mathbf{X}$ and the allele frequencies $\mathbf{p}$ are not known precisely. Instead, we use the expected genotypes given these probabilities (genotype dosages). To exclude large-effect SNP loci from human leukocyte antigen genes, SNPs on chromosome 6 in the 30–31 Mb region were masked and we verified that no significant associations were found in nearby regions after masking. Due to memory constraints, GCTA could not be run using a GRM of all 113,851 individuals at once. Instead, we divided all individuals into three equally sized batches, calculating the profile likelihood of $\alpha$ for each batch and using the sum of the resulting log likelihoods to compute the final likelihood curve.

Although our analysis does not require knowing all imputed genotypes precisely, we do assume that the genotype probabilities are well calibrated, i.e., that we are not overly confident in the imputation accuracy. Since imputation accuracy is difficult to assess if the number of minor alleles in the reference panel is very low, we only used SNP loci that had 5 or more minor alleles in the UK10K reference panel (MAF > 0.07%) in our main analysis. To further assess calibration of imputation noise, we compared the uncertainty implied by the genotype probabilities with an empirical assessment of imputation accuracy performed by the UK Biobank study (see URLs). Supplementary Figure 3 shows that imputation accuracy is significantly overestimated for SNPs of frequency 0.1% or less, which could potentially bias our results. However, repeating $\alpha$ estimation only using SNPs of MAF > 0.3% did not lead to significantly different results, implying that our results are not significantly affected (see Supplementary Table 3).

**Simulations.** Simulations were performed using genotype data from an $N = 5000$ random subset of the 113,851 unrelated British UK Biobank individuals. We used $M = 100,000$ consecutive SNPs from a 25 Mb block of chromosome 1. $N$ and $M$ where chosen to approximately match the power of the UK Biobank traits analyzed (which scales with $N/\sqrt{M}$; ref. [21]). As in the main analysis of UK Biobank traits, only SNPs with at least 5 minor alleles (MAF > 0.07%) in the UK10K reference panel were included. Phenotype values were generated using the linear model described in Eq. (4). The trait effect of the $j^{\mathrm{th}}$ SNP was drawn from $\beta_j | p_j, \mathrm{LLD}_j \sim N(0, \sigma_{g,\alpha}^2 \cdot [2p_j(1 - p_j)]^{\alpha} \cdot (1 + \tau^* \cdot \mathrm{LLD}_j))$, with $\tau^* = -0.3$ when simulating LD-dependent architectures[17], and $\tau^* = 0$ otherwise. The environmental noise variance was chosen such that the simulated trait had the desired heritability. In simulations with only 1% of SNPs causal, the causal SNPs were chosen at random. Imputation noise was introduced by randomly sampling the genotypes used to simulate phenotypes from imputed genotype probabilities, as reported by UK Biobank. In simulations without imputation noise, genotype dosages, i.e., the expected number of minor alleles, were used. In the inference procedure, we used genotype dosages in both types of simulations. In analyses reported in Supplementary Table 1, we used a larger set of SNPs (860,000) and individuals (15,000). As above, these values were chosen to approximately match the power of the UK Biobank traits analyzed; we note that chromosome 1 contains < 900,000 SNPs in our analysis.

Simulations to estimate $\alpha_{\mathrm{LDAK}}$ were performed using the same set of 5000 individuals and 100,000 SNP loci. Phenotype values were simulated as described above. $\alpha_{\mathrm{LDAK}}$ estimation was performed in the same way as in the previous set of simulations, only now using the LDAK software[19] to calculate the likelihood for a given $\alpha$ value instead of the GCTA software. This approach hence includes the LD

weights proposed by LDAK and is identical to their proposed approach for estimating $\alpha$, although, to enable a more accurate comparison, we used a finer set of tested $\alpha$ values ($\alpha' \in \{-1.00, -0.95,...,0.60\}$) than in their study ($\alpha' \in \{-1.25, -1.00, -0.75, -0.50, -0.25, 0.00, 0.25\}$). Due to computational constraints we did not use their workflow for imputed genotypes, but rather used the same hard-called genotypes for both phenotype simulations and estimation, an option available in LDAK.

When simulating trait values with $T_{sim} = 0\%$, $T_{sim} = 5\%$ and $T_{sim} = 10\%$, we used genome-wide SNPs (MAF > 0.07%; $M = 11,062,620$) and set the number of samples ($N = 113,660$), $\alpha = -0.45$ and $h^2 = 0.61$, to match UK Biobank height. We randomly selected 1% of SNPs to be causal. In simulations with $T_{sim} = 0\%$, causal effect sizes were sampled from $N\left(0, \sigma_{g,\alpha}^2 \cdot \left[2p_j\left(1-p_j\right)\right]^\alpha \cdot \left(1 + \tau^* \cdot LLD_j\right)\right)$. In simulations with $T_{sim} = 5\%$, causal effect sizes were sampled from $N\left(0, \sigma_{g,\alpha}^2 \cdot [2 \cdot 0.05(1-0.05)]^\alpha \cdot \left(1 + \tau^* \cdot LLD_j\right)\right)$ for all causal SNPs with MAF $\leq$ 5%, and accordingly in simulations with $T_{sim} = 10\%$.

**Correcting for bias in heritability estimation.** Heritability estimation methods based on standard restricted maximum likelihood (REML) estimation in a linear mixed model framework[16] require that all SNP effects are i.i.d. distributed in order to avoid biases. In the case of MAF-dependent SNP effects, this assumption is clearly broken. This issue has been addressed in previous work and several solutions to this problem have been suggested[15,19]. Here we show that knowing $\alpha$ for a given trait can provide another way to avoid heritability estimation biases due to MAF-dependent architectures. As previously stated, our model assumes $\mathbf{y} = \mathbf{X}\boldsymbol{\beta} + \boldsymbol{\varepsilon}$, with $\varepsilon_i \sim N(0, \sigma_\varepsilon^2)$ i.i.d. and $\boldsymbol{\beta}_j \sim N\left(0, \sigma_{g,\alpha}^2 \cdot \left[2p_j\left(1-p_j\right)\right]^\alpha\right)$. Here $\beta_j$ is the per-allele effect, the average effect on the phenotype of having one minor allele. However, one can define renormalized genotypes $\tilde{\mathbf{X}}$, with

$$\tilde{X}_{ij} = X_{ij} \cdot \left[2p_j\left(1-p_j\right)\right]^{\alpha/2}.$$

The per-normalized-allele effects are now $\tilde{\boldsymbol{\beta}}_j \sim N(0, \sigma_{g,\alpha}^2)$ i.i.d. in $\mathbf{y} = \tilde{\mathbf{X}}\tilde{\boldsymbol{\beta}} + \boldsymbol{\varepsilon}$. Since $\tilde{\beta}_j$ are now iid, $\sigma_{g,\alpha}^2$ and $\sigma_\varepsilon^2$ can now be estimated without bias from $\tilde{\mathbf{X}}$ and $\mathbf{y}$ using REML. The variance in the phenotype explained by $M$ SNPs can be calculated in the following way:

$$\sigma_g^2 = Var(\tilde{\mathbf{x}}\tilde{\boldsymbol{\beta}}) = \tilde{\boldsymbol{\beta}}^T Var(\tilde{\mathbf{x}})\tilde{\boldsymbol{\beta}} \approx E_{\hat{\beta}}(\tilde{\boldsymbol{\beta}}^T Var(\tilde{\mathbf{x}})\tilde{\boldsymbol{\beta}}) = \sum_{j=1}^M \sigma_{g,\alpha}^2 \left[2p_j\left(1-p_j\right)\right]^{1+\alpha}, \tag{9}$$

where $\tilde{\mathbf{x}}$ is a random renormalized genotype row vector. Here we used the fact that $(Var(\tilde{\mathbf{x}}))_{jj} = 2p_j\left(1-p_j\right)$ under Hardy-Weinberg equilibrium and cross terms cancel since $\tilde{\beta}_j$ are independent and mean zero. We define $A = \sum_{j=1}^M \left[2p_j\left(1-p_j\right)\right]^{1+\alpha}$, with the genetic variance $\sigma_g^2 = A \cdot \sigma_{g,\alpha}^2$. If $\alpha = -1$, as has been used in many previous methods[14,16], $A$ is simply equal to $M$.

In practice, heritability estimation was performed in the following way: the renormalized genotype matrix $\tilde{\mathbf{X}}$ was calculated using the $\hat{\alpha}$ as estimated from the data. From $\tilde{\mathbf{X}}$ and the phenotype vector, $\hat{\sigma}_{g,\alpha}^2$ and $\hat{\sigma}_\varepsilon^2$ were obtained using GCTA REML[37]. Our SNP heritability estimate $\hat{h}_{\alpha,noLD}^2$ is then defined as $\hat{A}\hat{\sigma}_{g,\alpha}^2 / (\hat{A}\hat{\sigma}_{g,\alpha}^2 + \hat{\sigma}_\varepsilon^2)$, with $\hat{A} = \sum_{j=1}^M \left[2p_j\left(1-p_j\right)\right]^{1+\hat{\alpha}}$. $\hat{h}_\alpha^2$ was calculated equivalently only now including previously described LD weights, i.e. we used $\left[2p_j\left(1-p_j\right)\right]^{\hat{\alpha}/2} \cdot \left(1 - 0.3 \cdot LLD_j\right)^{1/2}$ instead of $\left[2p_j\left(1-p_j\right)\right]^{\hat{\alpha}/2}$ when calculating $\tilde{\mathbf{X}}$ and $\hat{A}$.

**Phenotype selection and preprocessing.** In this analysis we investigated 25 highly heritable and polygenic human traits (see Table 2) from the UK Biobank study (see URLs). Specifically, we required a SNP heritability of 0.2 or more for quantitative traits and 0.1 or more for case-control traits (on the observed scale, see ref .[38]), as well as at least 50% of the 113,851 British ancestry individuals to be phenotyped. We also removed phenotypes for which the top 10 SNPs explained 10% or more of the trait variance, so as to avoid $\alpha$ estimates that are dominated by a few top SNPs, as our goal is to study polygenic architectures. (Only one trait, mean platelet volume, was removed due to this restriction.) The 25 traits that we chose include 21 quantitative traits and 4 case-control traits. Eleven of the quantitative traits are blood cell traits, whereas the remaining 14 include a wider range of physiological measurements and diseases. Since the number of available blood cell traits was large and many of them were highly correlated, we additionally required blood cell traits to have a pairwise phenotypic correlation of $r^2 < 0.5$, removing the less heritable trait for any correlated pair.

For each trait, phenotype values had outliers removed and fixed effects were regressed out. Specifically, phenotype values 4 or more standard deviations away from the mean (or similarly extreme outliers for skewed distributions) were removed from the analysis. Sex and ten principal components of the GRM were included as fixed effects for all traits, with additional trait specific covariates also

included for some traits (see Supplementary Table 10). All trait values were then rank-based inverse normal transformed before being analyzed.

**Inferring fitness-trait coupling and selection parameters.** We aimed to use the frequency dependence of SNP effects to draw conclusions about the fitness effects of SNPs, as well as the coupling between fitness and the target trait effects. Let $\beta^2|p$ be the squared trait effect size of a SNP given its MAF $p$, and $s$ the fitness effect of the SNP, which is here assumed to be deleterious or neutral. From the law of total expectation it follows that $E(\beta^2|p) = E(E(\beta^2|s, p)|p)$. The main assumption of this analysis is that, at a given selection coefficient, the effect size of the SNP is independent of its frequency, i.e., $E(\beta^2|s, p) = E(\beta^2|s)$. This is equivalent to the statement that the frequency dynamics of a SNP is influenced by $\beta^2$ only through $s$. We then use the model of Eyre-Walker[9], where the absolute value of $\beta$ is proportional to $s^\tau(1 + \varepsilon)$, with $\varepsilon \sim N(0, \sigma^2)$ and $\tau$ indicating how strongly $\beta$ depends on $s$. It follows that $E(\beta^2|s) \propto s^{2\tau}$ and from above, for some constant $c$,

$$E\left(\boldsymbol{\beta}^2|s\right) = c \cdot E\left(s^{2\tau}|p\right) \tag{10}$$

Given a positive $\tau$, this equation shows that increased average effects of lower-frequency SNPs requires lower-frequency SNPs having increased $s$ and hence implies significant negative selection. Some previous analyses[4,6,32] have argued that in the absence of selection, SNPs of MAF ranges of equal width (e.g., 5–10% and 10–15%) are expected to explain an equal fraction of heritability. However, even in the absence of selection, population expansion can lead to excess rare variants, leading to increased rare variant heritability[43]. Increased rare variant heritability is therefore not necessarily a sign of selection.

Assuming we know $\tau$ and the joint distribution of $s$ and $p$, $E(\beta^2|p)$ can be derived from Eq. (10). We simulated samples of this distribution using the evolutionary forward simulation framework SLiM2 (ref. [25]). Simulations were run with a European demographic model inferred by ref. [26], a burn-in of 3880 generations before the bottleneck, a mutation rate of $2 \times 10^{-8}$ per base pair per individual per generation[44], and a recombination rate of $10^{-8}$ per base pair per individual per generation[45]. These simulations also require assumptions about the distribution of fitness effects (DFE), i.e., the distribution of $s$ for de novo mutations, but the DFE for genome-wide SNPs in humans is currently not known. We assumed a gamma distributed DFE, using a plausible range of average fitness effects, $\bar{s} \in \{10^{-3}, 10^{-4}\}$ (note that $\bar{s} \leq 10^{-5}$ would lead to $T$ close to or greater than 0.5 and hence no or almost no frequency dependence; see Supplementary Figure 4, panel (e)), and shape parameters of $k = 0.125$ and 0.25 which includes the range of plausible values derived by ref. [24]. In the analyses reported in Supplementary Figure 5, we considered a broader range of shape parameters ($k = 1, 0.5, 0.25, 0.125, 0.0625$). Although the gamma distribution does not include variants with exactly zero effect size, $k$ effectively parametrizes the polygenicity of fitness effects. For example, at $k = 0.0625$, the mean SNP fitness effect is almost 7000 times larger than the median SNP fitness effect, indicating that fitness effects are extremely sparse. At $k = 1$, the mean is only 1.4 times larger than the median, indicating that fitness effects are extremely polygenic. Since SNPs with negligible fitness effects also have negligible effects on the target trait under the Eyre-Walker model[9], the shape parameter $k$ determines the polygenicity of both fitness and the target trait.

For each choice of DFE we simulated 25 independent replicates over a 4 Mb block each, for a total of 100 Mb with each DFE. In all simulations the Eyre-Walker noise parameter, $\sigma^2$, was set to zero. This parameter does not change SNP effects on average and is therefore negligible in the limit of large SNP numbers. This was also noted in original analysis by ref. [9].

In the absence of LD between selected SNPs and assuming a constant effective population size $N_e$, $E(\beta^2|p)$ can also be derived analytically. Under these assumptions and assuming mutation rate per base pair $\mu \ll 1/N_e$ (ref. [44]), it is known that $P(p|s) \propto [p(1-p)]^{-1}e^{-4N_e sp}$ (ref. [46]). Given $s$ is drawn from a gamma distribution with mean $\bar{s}$ and shape parameter $k$, we obtain

$$E(\boldsymbol{\beta}^2|p) = c \cdot E(s^{2\tau}|p) = c \cdot \frac{\int_0^\infty s^{2\tau} P(p|s)P(s)ds}{\int_0^\infty P(p|s)P(s)ds}$$
$$\approx c \cdot \frac{\Gamma(2\tau+k)}{\Gamma(k)}(4N_e)^{-2\tau}\left[p + \frac{k}{4N_e\bar{s}}\right]^{-2\tau} \tag{11}$$

This result shows that for $p \ll \frac{k}{4N_e\bar{s}}$, $E(\beta^2|p)$ is constant, whereas for $p \gg \frac{k}{4N_e\bar{s}}$ it falls off as $p^{-2\tau}$. We note that these calculations imply $\alpha \approx -2\tau$, whereas $\alpha$ is significantly less negative in simulations (see Fig. 3), with the difference likely being due to LD between SNPs with different selection coefficients (see Discussion). For simplicity, we have here assumed that $p$ is the derived allele frequency – if $p$ is the minor allele frequency (MAF), results are generally similar, although they differ for very common SNPs. Specifically, when using MAF, $E(\beta^2|p) \propto ([p + T]^{-2\tau-k} + [1 - p + T]^{-2\tau-k})/([p + T]^{-k} + [1-p + T]^{-k})$, with threshold frequency $T = \frac{k}{4N_e\bar{s}}$ (see caption of Supplementary Figure 6). We show in Supplementary Figure 6 that our analytical result using MAF matches the $\alpha$ model more closely (though not perfectly), as the correction terms for very common SNPs match the $(1-p)$ factor in the $E(\beta^2|p) \propto [p(1-p)]^\alpha$ model.

When fitting $\alpha$ to SNP effects from a simulation with a given $\bar{s}$, $k$ and $\tau$ in Fig. 3, we only used SNPs with frequency above $\frac{k}{4N_e\bar{s}}$. $(\hat{c}', \hat{\alpha})$ is calculated by minimizing the squared deviation between $c' \cdot [p(1-p)]^\alpha$ and the simulated SNP effects

summed over all SNPs from 25 independent simulations. Error bars were obtained by bootstrap resampling of these 25 simulations. The proportionality constant in Eq. (10) does not affect $\hat{\alpha}$ and was set to $c = 1$. When estimating $\tau$ from $\hat{\alpha}$ of a given trait, we assumed a flat prior on $\alpha$ over $[-1,0]$ and on $\tau$ over $[0,1]$, in which case

$$P(\tau|\text{data}) \propto \int_{-1}^{0} P(\alpha|\tau)P(\alpha|\text{data})\mathrm{d}\alpha.$$ Here, $P(\alpha|\text{data})$ is proportional to the

calculated profile likelihood and $P(\alpha|\tau)$ is based on estimates and error bars displayed in Fig. 3, assuming equal probability for $\bar{s} = 10^{-3}$ and $\bar{s} = 10^{-4}$, and $k = 0.25$. Using $k = 0.125$ lead to similar results, e.g., $\alpha = -0.38$ then corresponds to $\tau \in [0.33, 0.43]$ instead of $\tau \in [0.32, 0.48]$ for $k = 0.25$.

**URLs**. Open-source software package implementing our method, https://github.com/arminschoech/GRM-MAF-LD; UK Biobank website, http://www.ukbiobank.ac.uk/; BGEN file format, http://www.well.ox.ac.uk/~gav/bgen_format/; UK Biobank genotype imputation manual, http://www.ukbiobank.ac.uk/wp-content/uploads/2014/04/imputation_documentation_May2015.pdf

**Code availability**. Source code of the GRM-MAF-LD software developed for this analysis is publicly available at https://github.com/arminschoech/GRM-MAF-LD.

**Reporting summary**. Further information on experimental design is available in the Nature Research Reporting Summary linked to this article.

## Data availability

This work used data from the UK Biobank study (http://www.ukbiobank.ac.uk/). The data is not publicly available but researchers can apply to use the resource.

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

## Acknowledgements

We are grateful to Ivana Cvijović, Kevin Galinsky, Alexander Gusev, Benjamin Neale, Nick Patterson and David Reich for helpful discussions. This research was funded by NIH grants R01 MH101244 and U01 HG009088 and by a Boehringer Ingelheim Fonds fellowship. This research was conducted using the UK Biobank Resource under Application Number 16549. Computational analyses were performed on the Orchestra High-Performance Compute Cluster at Harvard Medical School.

## Author contributions

A.P.S., P.-R.L., L.O., D.J.B., P.F.P., H.K.F., S.R.S. and A.L.P. conceived and designed the analysis. A.P.S., D.J. and S.G. performed the analysis. A.P.S., D.J., S.R.S. and A.L.P. analyzed the data. A.P.S. and A.L.P. wrote the manuscript with input from all authors.

## Additional information

**Competing interests:** The authors declare no competing interests.

