## [Peer Review File · Nature Communications]

REVIEWERS' COMMENTS:

Reviewer #1 (Remarks to the Author):

The authors have faced multiple rounds of revisions at Nature Genetics, where relevant and interesting points were raised by all reviewers. The authors have gone to great lengths to address all of the comments raised at each stage and I believe that the manuscript has been greatly strengthened as a result of these efforts.

Unfortunately, it was deemed that the manuscript was more suitable for a less impactful journal. However, I wholeheartedly support acceptance of this piece of work in Nature Communications and I think it is a relevant and timely contribution to the literature that supports the conclusions of a complementary paper that happened to be submitted to Nature Genetics at the same time, which was accepted. The authors have addressed all of my previous concerns and, so far as I can see, all of the comments of the other reviewers very comprehensively. I find the results to be robust, the manuscript to be well-written, and the various issues and caveats to be much better described than the already published complimentary paper. Additionally the population genetics theory and simulation work are also, in my opinion, superior to the previous work.

Reviewer #1 (Remarks to the Author):

The authors have faced multiple rounds of revisions at Nature Genetics, where relevant and interesting points were raised by all reviewers. The authors have gone to great lengths to address all of the comments raised at each stage and I believe that the manuscript has been greatly strengthened as a result of these efforts.

Unfortunately, it was deemed that the manuscript was more suitable for a less impactful journal. However, I wholeheartedly support acceptance of this piece of work in Nature Communications and I think it is a relevant and timely contribution to the literature that supports the conclusions of a complementary paper that happened to be submitted to Nature Genetics at the same time, which was accepted. The authors have addressed all of my previous concerns and, so far as I can see, all of the comments of the other reviewers very comprehensively. I find the results to be robust, the manuscript to be well-written, and the various issues and caveats to be much better described than the already published complimentary paper. Additionally the population genetics theory and simulation work are also, in my opinion, superior to the previous work.

We thank the reviewer for the helpful feedback during the review process and for supporting publication of the manuscript in Nature Communications.